# Evaluation of Chemical Composition of *Miscanthus* × *giganteus* Raised in Different Climate Regions in Russia

**DOI:** 10.3390/plants11202791

**Published:** 2022-10-21

**Authors:** Yulia A. Gismatulina, Vera V. Budaeva, Aleksey N. Kortusov, Ekaterina I. Kashcheyeva, Evgenia K. Gladysheva, Galina F. Mironova, Ekaterina A. Skiba, Nadezhda A. Shavyrkina, Anna A. Korchagina, Vladimir N. Zolotukhin, Gennady V. Sakovich

**Affiliations:** Bioconversion Laboratory, Institute for Problems of Chemical and Energetic Technologies, Siberian Branch of the Russian Academy of Sciences (IPCET SB RAS), Biysk 659322, Russia

**Keywords:** *Miscanthus*, chemical composition, cellulose, lignin, pentosans, different climate regions

## Abstract

Lignocellulosic biomass is of great interest as an alternative energy resource because it offers a range of merits. *Miscanthus × giganteus* is a lignocellulosic feedstock of special interest, as it combines a high biomass productivity with a low environmental impact, including CO_2_ emission control. The chemical composition of lignocellulose determines the application potential for efficient industrial processing. Here, we compiled a sample collection of *Miscanthus × giganteus* that had been cultivated in different climate regions between 2019 and 2021. The chemical composition was quantified by the conventional wet methods. The findings were compared with each other and with the known data. Starting as soon as the first vegetation year, *Miscanthus* was shown to feature the following chemical composition: 43.2–55.5% cellulose content, 17.1–25.1% acid-insoluble lignin content, 17.9–22.9% pentosan content, 0.90–2.95% ash content, and 0.3–1.2% extractives. The habitat and the surrounding environment were discovered herein to affect the chemical composition of *Miscanthus*. The stem part of *Miscanthus* was found to be richer in cellulose than the leaf (48.4–54.9% vs. 47.2–48.9%, respectively), regardless of the planation age and habitat. The obtained findings broaden the investigative geography of the chemical composition of *Miscanthus* and corroborate the high value of *Miscanthus* for industrial conversion thereof into cellulosic products worldwide.

## 1. Introduction

Fossil fuel is a non-renewable energy source having a crucial importance for global development [1], and it is likely that it will be depleted within the next 40–50 years [2]. It is essential to advance alternative energy sources in order to replace fossil fuel resources, mitigate greenhouse gas emissions and abate the anthropogenic burden on the environment [2,3]. Lignocellulosic biomass is of great interest as an alternative energy resource because of the significant benefits it offers: great diversity, availability, carbon neutrality and low cost as compared to fossil fuels [2,4,5]. As of today, lignocellulosic biomass is estimated at almost 25% of the global energy supply [6].

Perennial C4 plants are considered to be an especially promising alternative feedstock because of their higher photosynthetic capacity, high yield and productive utilization of nitrogen and water when compared to C3 plants [3,7]. These peculiarities allow the perennial grasses, particularly *Miscanthus*, to attain to substantial yields [8] even if they are raised in marginal and degraded lands [7,9,10], thereby exerting a positive habitat-forming impact on them. That said, *Miscanthus* is reckoned as one of the most promising because it is capable of utilizing the ambient resources more efficiently than the other C4 plants [10].

The *Miscanthus* cultivation is already beneficial by itself for environmental restoration, for example, CO_2_ fixation [11] and remediation [12,13]. The use of *Miscanthus* as the carbon resource can replace fossil fuels, with no serious damage to the environment [2]. In China, *Miscanthus* has already become in-demand for bioenergy development [14]. In the UK, *Miscanthus* is also touted as a popular bioenergy feedstock because of high yields (8–32 ton/ha) and a high energy output (140–560 GJ/ha) compared to the other raw materials [9,14]. It is believed that *Miscanthus* is able to shorten and, in perspective, replace wood in the industry if more eco-benign (green) technologies are employed, particularly without sulfur and chlorine chemicals and hence with a minimum negative impact on the environment [15]. Baxter et al. [16] noted that even the combustion of *Miscanthus* biomass releases harmful agents no more than those absorbed during the growth, making the closed carbon cycle feasible. Kowalczyk-Juśko et al. [17] reported the broadest variability of biochemical conversion of *Miscanthus* into a range of valuable products.

The following Miscanthus varieties are being studied most: Miscanthus sinensis, *Miscanthus* × *giganteus*, Miscanthus sacchariflorus and Miscanthus floridulus.

In the present study, we examined *Miscanthus × giganteus*, a perennial cereal crop with a biomass yield of up to 40 ton/ha annually [7,10] over the span of 18–25 years, having a high potential for the greenhouse gas mitigation through carbon fixation into the soil [18]. The advantage of *Miscanthus × giganteus* is that it is able to sequestrate twice more carbon than *Miscanthus sinensis* [19].

The cultivation of *Miscanthus × giganteus* in Russia is discussed as compared with other countries (edaphoclimatic conditions, yield capacity, harvesting time and constituent composition of ash) in a series of studies [20,21,22,23], but there is no data on the chemical composition (contents of polymers) of those harvests.

The chemical composition of any lignocellulosic feedstock needs to be evaluated to identify whether the feedstock has a value for converting the same into products in demand [1]. The chemical composition of *Miscanthus × giganteus* is currently being studied extensively [24,25,26,27,28,29,30,31,32,33]. Despite quantification methods for the *Miscanthus* chemical composition being diverse, they are more alike than distinct, making the comparison between the reported data possible [1]. Studies on chemical composition measurement and variations after chemical modifications are being pursued [34,35,36]. There are known studies on the chemical composition of *M. saccharif lorus*, *M. Sinensis* and *M. Purpurascens* raised in the continental climate of West Siberia (Russia) [37].The relationships between the chemical composition of *Miscanthus* and species/variety, plantation age, climatic conditions, seasonal variations and harvest time are being examined [10]. Studies are being performed on the chemical composition subject to the biomass harvesting time: early (fall) and late (spring) [7,10,18,25,26,27,28]. Wahid et al. [28] observed no changes in the chemical composition of *Miscanthus × giganteus*, which is more likely due to the fact that it had already achieved an optimum maturity at early harvesting. Cellulose was shown to prevail in the spring harvest [7,25,26,27], which is the best for further processing of *Miscanthus*. The spring harvesting is also preferable because of a low biomass moisture and more complete transfer of nutrients from leaves and stems to rhizomes for storage and utilization in the next season [10,18,25,26,27], favorably affecting the soil quality [38]. The effects of plantation age on the biomass yield [7,28,39] and chemical composition of *Miscanthus* are being explored [24,40]. There are single studies on the chemical composition quantification of different morphological portions of *Miscanthus* [28,29,31,41].

Despite *Miscanthus* being evidently promising for carbon footprint mitigation and/or for its conversion into valuable products, there are no studies on the chemical composition of *Miscanthus × giganteus* raised in three different climate regions. The present study would expand the knowledge of the chemical composition of *Miscanthus × giganteus* as a function of the habitat, plantation age and morphological part of the plant.

The present paper aimed to evaluate the chemical composition of *Miscanthus × giganteus* subject to the climate region of cultivation and to the morphological part of the plant (leaf and stem).

## 2. Materials and Methods

### 2.1. Miscanthus Samples

*Miscanthus × giganteus* samples were provided by the farmers from seven different plantations with vegetation years of 2019–2021 (Table 1). The plantations are all located within Russia in the following cities grouped by the climate regions: Kaluga, Moscow, Bryansk, Kaliningrad and Penza (the temperate continental climate region), Novosibirsk (the continental climate region), and Irkutsk (the severely continental climate region).

The biomass was harvested in spring next year before the new vegetation season started, as recommended in [10,25,26,27]. The whole aboveground portion of *Miscanthus* (cut 10–15 cm above the ground) was used for the chemical composition quantification. The *Miscanthus* biomass was composed chiefly of stems, as the leaves are less able to withstand wind and frost. The *Miscanthus* biomass was ground for chemical composition quantification and, if necessary, air-dried to a moisture of, at most, 8%. 

We determined the chemical composition of eleven *Miscanthus* samples from the whole plant having the following habitats and plantation ages: Kaluga, aged 1, 4 and 5 years; Moscow, aged 3 and 7 years; Bryansk, aged 1 and 2 years; Kaliningrad, aged 2 years; Penza, aged 8 years; Novosibirsk, aged 1 year; and Irkutsk, aged 1 year. The data on the harvesting year, farmers and yield capacity (calculated), sample weight and climatic conditions for cultivation of *Miscanthus × giganteus* differing in plantation age and habitats (Russia) are outlined in Table 1. We also quantified chemical compositions of four *Miscanthus* samples from different morphological parts, the leaf and stem, of the plants having the following habitats and plantation ages: Kaluga, aged 1 and 5 years; Kaliningrad, aged 2 years; and Moscow, aged 3 years. These samples were provided by the farmers to us. 

### 2.2. Chemical Composition of Miscanthus

The standard analytical techniques, also known as the wet ones, which rely on feedstock fractionation and are most commonly used for cellulosic biomass were employed for the quantification of chemical constituents of *Miscanthus* [1].

The cellulose content was measured by the Kürschner method by extracting a weighed portion of *Miscanthus* with mixed alcohol/nitric acid in a ratio of 4:1 [5,42]. The acid-insoluble lignin content was determined using 72% sulfuric acid as per the TAPPI standard [43]. Pentosans were quantified by transforming the same in boiling 13 wt.% HCl solution into furfural which was collected in the distillate and determined on a xylose-calibrated UNICO UV-2804 spectrophotometer (United Products & Instruments, Dayton, NJ, USA) at a 630 nm wavelength using orcinol–ferric chloride [44]. The ash content was quantified by the TAPPI standard [45]. The extractives were quantified by the TAPPI standard [46] after successive extraction with methylene chloride in a Soxhlet extractor. Moisture was analyzed on an OHAUS MB-25 moisture analyzer (Parsippany, NJ, USA). All the experiments were performed in triplicate and the data were expressed as the means. 

The analyses were done using equipment provided by the Biysk Regional Center for Shared Use of Scientific Equipment of the SB RAS (IPCET SB RAS, Biysk city, Russia).

## 3. Results and Discussion

The chemical composition is the most important indicator to evaluate if a plant feedstock has the potential for efficient industrial processing. The *Miscanthus* cell wall consists mainly of the polymers such as cellulose, hemicelluloses and lignin [10]. There are guidelines on how to assess the biomass quality after a plant growth period of at least 2–3 years [10]. Here, we examined the biomass of *Miscanthus* from different-aged plantations, including one-year-old plants. Table 2 summarizes the chemical compositions of *Miscanthus × giganteus* differing in plantation age and habitats.

It follows from the tabulated data (Table 2) that, starting as soon as the first vegetation year, *Miscanthus* exhibits the following chemical composition: 43.2–55.5% cellulose content, 17.1–25.1% acid-insoluble lignin content, 17.9–22.9% pentosan content, 0.90–2.95% ash content and 0.3–1.2% extractives. 

By comparing the chemical compositions of *Miscanthus* from different climate regions, it can be noted that *Miscanthus* plants from the continental (Novosibirsk) and severely continental (Irkutsk) climate regions are similar in biomass indicators. It was found by comparing these values with the chemical composition of *Miscanthus* plants from the same-age plantations growing in the temperate continental climate that the contents of cellulose, lignin and pentosans are higher by 2.4–4.6%, 1.6–4.4% and 2.0–5.0% in the latter climate, respectively, with the extractives content being almost similar and the ash content being 0.85–2.05% lower.

It should be noted that our data obtained for the continental climate (Novosibirsk) can be compared to those of chemical compositions of three *Miscanthus* species *Miscanthus sacchariflorus*, *Miscanthus Sinensis* and *Miscanthus Purpurascens* cultivated under the same climatic conditions [37]. It is obvious that the reason behind the low cellulose content of *Miscanthus × giganteus* sample (Table 2) is due to the plantation age (one year old) because all the three *Miscanthus* species taken for comparison were raised on plantations aged 5 years.

Because there are no publications on chemical composition measurement results for the Russian *Miscanthus × giganteus* varieties, we compared our findings with the international studies from the UK, the USA, Portugal, the Netherlands, Germany, France, Greece, Ukraine, Belgium, Korea and South Ireland. These countries are characterized by a milder climate, particularly by no high-negative temperatures.

Our findings on the chemical composition of *Miscanthus*
*×* *giganteus* are in agreement with the other studies: 32.7–52.9% cellulose content, 7.6–33.0% lignin content and 17.1–33.8% hemicellulose content [15,24,25,26,27,28,47,48]. However, the lignin content of the Russian *Miscanthus* is characterized by a narrower range of 17.1–25.1% when compared to the international results, suggestive of the impact of the habitat on the chemical composition of biomass [26,27,40]. The main peculiar feature of *Miscanthus × giganteus* raised in Russia is that it is capable of growing in the temperate and severely continental climates, with a high-efficiency productivity of biomass having a cellulose content as high as 55.5%. Such a high cellulose measure is commensurate with and, in some instances, superior to those of cellulose from other countries (41.8% in Greece, 37.8% in France, 42.3% in Germany and 49.5% in the UK) [26] located in the similar temperate climate zone and even in the warmer subtropical zone. In the study by Schläfle et al. [48], *Miscanthus × giganteus* raised in the moderate climate of Germany had the following chemical composition: 49.4% cellulose content, 27.7% acid-insoluble lignin content, 19.9% hemicellulose and 1.21% ash content, in a good agreement with the findings from the present study. Vanderghem et al. [47] reported chemical compositions of *Miscanthus × giganteus* raised in Belgium in the moderate marine climate with a mild winter and cool summer as: 48.4% cellulose, 23.0% acid-insoluble lignin, 17.6% pentosans and 2.4% ash content, which is also consistent with the present study results.

The maximum cellulose content was detected in the biomass of *Miscanthus* from the oldest 8-year-old plantation, which is in good agreement with the results from the other studies [40] in which a tendency of the increase in the cellulose content was noted for three *Miscanthus* species according to the plantation age. The same tendency was observed for *Miscanthus* raised on the territorially similar plantations but of different ages: a cellulose content increased from 47.8% (1 year) to 49.4% (4 years) and further to 50.2% (5 years) in Kaluga city, while the cellulose content rose from 46.8% (1 year) to 50.4% (2 years) in Bryansk city. Such a tendency was not noted for the Moscow plantation (50.1% cellulose for the plantations aged 3 and 7 years old), which can be due to the fact the highest increment in the cellulose content is observed exactly in the initial life years of the plantation [40], i.e., the cellulose content of the plant from the 3-year-old plantation almost achieved its ultimate level and no further increment was noted. By the example of the continental climate, without being bound to any particular city, one can observe a tendency of an increase in the cellulose content from 46.8–47.8% for 1-year-old plants to 50.4–53.5% for 2-year-old plants. No relationships for the measures of pentosans, lignin, ash content and extractives were established. The data obtained by other researchers on this matter are somewhat controversial. In the UK, three harvests (from plantations aged 2, 3 and 4 years) were found to have no considerable changes in the chemical composition of 244 *Miscanthus* genotypes, depending on the plant age, except for the ash content [24]. Weijde et al. [40] showed that the biomass of *Miscanthus* from the Ukrainian, Germany and Netherlands plantations was noted to increase in the cellulose content for the initial three vegetation years, which is in line with our findings.

By comparing the chemical composition of *Miscanthus* from plantations in five cities located in the temperate continental climate, the highest cellulose contents were detected for the two geographical locations: 53.5% in Kaliningrad and 55.5% in Penza; the other three cities (Kaluga, Moscow and Bryansk) are characterized by close cellulose contents ranging from 49.4% to 50.4%, starting from the second vegetation year. The similar values for the latter three geographical locations are explained by the cities being territorially close to each other and hence having identical climatic conditions. The high contents of cellulose (53.5%) and lignin (25.1%) in the biomass of *Miscanthus* from Kaliningrad are most likely due to this plant being territorially remote from the preceding three samples, namely, due to the plant being situated at the interface of the marine climate and the temperate continental climate and hence due to a milder climate with favorable humidity and daily average temperatures in summer and winter. The second city that is distant from Kaluga, Moscow and Bryansk is Penza in which the biomass exhibits a maximum cellulose content of 55.5% and a maximum ash content of 2.63%. Despite Penza being distant from Moscow, these are very alike in the climate, but the climate in Penza is more continental and arid. Because of the Penza *Miscanthus* biomass having an enhanced ash content, it can be inferred that the soil of that plantation is distinct from the other plantations, which could help the biomass to achieve such a high cellulose content. This considered, such a high cellulose content is due to the plantation being 8 years old [40].

However, despite the difference in the chemical compositions according to the plantation age, growth conditions and habitats, a fundamental pattern is observed regarding the contents of cellulose, lignin, pentosans, ash and extractives in the leaf and stem (Figure 1). 

It follows from Figure 1 that the stem contains most of cellulose (48.4–54.9% vs. 47.2–48.9%) and lignin (23.0–26.3% vs. 18.7–20.4%), while the leaf contains chiefly the other non-cellulosic constituents, more specifically ash (3.95–7.79% vs. 0.93–1.91%), pentosans (22.2–24.4% vs. 19.1–22.1%) and extractives (1.2–1.9% vs. 0.4–0.9%). The difference in the chemical composition is attributed to different metabolic mechanisms of the processes occurring in the leaves and stems. In particular, the stiffening of the stem compared to the flexible leaf is due to the higher lignin content [10]. The ash content of the leaf is 2.1–8.4 times higher than that of the stem, which is due to the leaves being richer in minerals, and is in agreement with the data reported in [31]. These comparison results allow for the conclusion that the *Miscanthus* stem is characterized by a higher cellulose content, irrespective of the habitat and plantation age.

It was found in Germany [31], as was in our study, that cellulose (50.0–50.5% vs. 44.8–45.0%) was concentrated in the *Miscanthus* stem, while pentosans and ash were concentrated in the leaf (28.4–29.5% and 4.53–6.82% vs. 26.2–27.4% and 2.50–3.07%, respectively), with no considerable differences in the lignin content (25.3–26.0%) detected in the leaf and stem. In Korea, located in the temperate climate region, the stems of three *Miscanthus* species were found to have a higher content of cellulose than the leaves, whereas the lignin content did not differ greatly between the leaves and stems [29]. It was discovered in South Ireland [28], as was in our study, that cellulose and lignin were concentrated in the stem of *Miscanthus × giganteus* (52.5% and 14.7% vs. 35.5% and 8.0%, respectively), while pentosans and ash were concentrated in the leaf (31.0% and 5.8% vs. 19.7% and 3.2%, respectively). Even though the tendency of the three basic constituents concentrated in the leaf and stem persists, the chemical composition of *Miscanthus* from South Ireland differs appreciably from our findings: a high ash content of 3.2–5.8% in the leaf and stem and a low cellulose content (35.5%) and a high pentosan content (31.0%) in the leaf, which is explained by the different climatic features of the countries, namely by a milder moist oceanic climate of South Ireland. In the USA (Iowa) [41], cellulose and lignin were also concentrated in the stem (41.6% and 25.6% vs. 33.2% and 24.1%), while pentosans in the leaf (17.7% vs. 17.1%). The pattern of the distribution of the basic constituents within the leaf and stem is observable again, but in particular, the leaf is much lower in cellulose content, which is due to the climatic features of the growth region, namely due to the continental climate with a hot arid summer and cold winter. 

Our study demonstrated that the habitat and climatic peculiarities have an impact on the chemical composition of *Miscanthus*. A more detailed evaluation of the chemical composition of *Miscanthus × giganteus* requires that different-age *Miscanthus* should be further harvested from the same plantation, which will reveal new patterns and relationships, and the creation of a Russia-wide chemical composition database should be continued. 

The low content of lignin in *Miscanthus × giganteus* and its conceptual distinction from wood lignin [18], along with the high cellulose content (up to 55.5%), allow this crop to be reckoned as a feedstock for the manufacture of an array of valuable products. Moreover, based on the chemical composition measurements of *Miscanthus* that is a new cellulosic source for Russia, it can be concluded that *Miscanthus* has a lead position among other non-woody species, as reported likewise in [36,49,50,51,52,53,54].

Thus, *Miscanthus **×** giganteus* can be esteemed as a crop of high importance for the national ecology and industries because this plant forms “carbon quotes” for greenhouse gas emissions and is able to compete with fossil energy sources. 

## 4. Conclusions

*Miscanthus* was discovered to exhibit the following chemical composition, starting from the first vegetation year: 43.2–55.5% cellulose content, 17.1–25.1% acid-insoluble lignin content, 17.9–22.9% pentosan content, 0.90–2.95% ash content and 0.3–1.2% extractives. The habitat and the surrounding environment were found to influence the chemical composition of *Miscanthus*. *Miscanthus* plants raised at the interface of marine and temperate continental regions with favorable humidity and daily average temperature in summer and autumn compare favorably with the other samples in terms of the cellulose content (53.5%). *Miscanthus* from the 8-year-old plantation has the maximum cellulose content (55.5%), as evidenced by the cellulose increment as the plantation age was advancing. The stem part of *Miscanthus* offers a key merit: the stem is richer in cellulose than the leaf (48.4–54.9% vs. 47.2–48.9%, respectively). This fundamental phenomenon gives a rationale for harvesting *Miscanthus* in spring in the regions with dry winter, when *Miscanthus* naturally drops off the leaf and governs, chiefly, the use of the stem part when processed into cellulose. 

The findings obtained herein broaden the geographic boundaries of the environmental triumph of *Miscanthus*, providing mankind with a raw material base for the manufacture of fuel and chemicals at present and in the future.

The findings suggest that it will become possible in the nearest decade to perform a screening of chemical compositions of *Miscanthus × giganteus* raised on plantations of the same age in different regions of the country. 

## Figures and Tables

**Figure 1 plants-11-02791-f001:**
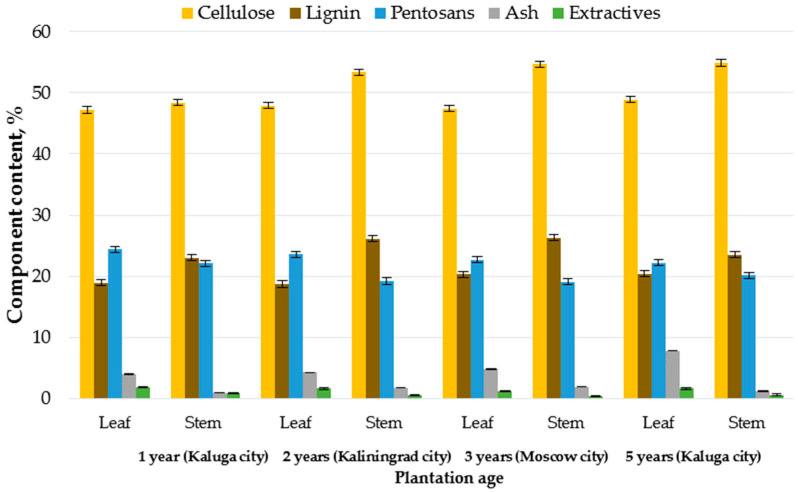
The chemical composition of the leaf and stem of *Miscanthus **×** giganteus* raised in Kaluga (plantation aged 1 year), Kaliningrad (plantation aged 2 years), Moscow (plantation aged 3 years) and Kaluga (plantation aged 5 years).

**Table 1 plants-11-02791-t001:** Data on harvesting year, farmers and yield capacity (calculated), sample weight and climatic conditions for cultivation of *Miscanthus × giganteus* differing in plantation age and habitat (Russia).

Plantation Age/Habitat/Harvesting Year	Farmer	Yield Capacity, t/ha	Sample Weight, kg	Annual Means	
				T, °C	Rainfalls, mm
1 year old, Kaluga, 2020	OOO Re:forma	2.5	7.6	+5.6	636
4 years old, Kaluga, 2021	OOO Re:forma	21.5	4.0	+5.6	636
5 years old, Kaluga, 2022	OOO Re:forma	22.0	4.5	+5.6	636
3 years old, Moscow, 2020	OOO Master Brand	14.5	11.0	+5.8	739
7 years old, Moscow, 2020	OOO Master Brand	19.0	6.8	+6.0	825
1 year old, Bryansk, 2021	Farm Household Savchenko V.V.	2.9	2.5	+6.1	671
2 years old, Bryansk, 2022	Farm Household Savchenko V.V.	10.0	3.0	+6.1	671
2 years old, Kaliningrad, 2021	OOO Kalagra Farm	12.0	2.4	+7.9	750
8 years old, Penza, 2022	Penza State Agrarian University	22.0	15.0	+5.2	521
1 year old, Novosibirsk, 2020	Siberian Research Institute of Plant Cultivation and Breeding	2.1	1.0	+2.6	437
1 year old, Irkutsk, 2022	OOO Sibgiprobum	2.0	1.5	+1.0	472

**Table 2 plants-11-02791-t002:** Chemical composition of *Miscanthus × giganteus* differing in plantation age and habitat (Russia).

Plantation Age/Habitat/Harvesting Year	Component Content, %				
	Cellulose	Lignin	Pentosans	Ash	Extractives
1 year old, Kaluga, 2020	47.8 ± 0.5	21.5 ± 0.5	22.9 ± 0.5	0.90 ± 0.05	1.2 ± 0.1
4 years old, Kaluga, 2021	49.4 ± 0.5	20.6 ± 0.5	22.3 ± 0.5	1.45 ± 0.05	0.5 ± 0.1
5 years old, Kaluga, 2022	50.2 ± 0.5	19.6 ± 0.5	20.4 ± 0.5	1.58 ± 0.05	0.6 ± 0.1
3 years old, Moscow, 2020	50.1 ± 0.5	21.7 ± 0.5	21.0 ± 0.5	1.55 ± 0.05	0.8 ± 0.1
7 years old, Moscow, 2020	50.1 ± 0.5	25.0 ± 0.5	21.7 ± 0.5	0.96 ± 0.05	0.7 ± 0.1
1 year old, Bryansk, 2021	46.8 ± 0.5	21.3 ± 0.5	22.2 ± 0.5	1.76 ± 0.05	0.5 ± 0.1
2 years old, Bryansk, 2022	50.4 ± 0.5	18.5 ± 0.5	22.6 ± 0.5	1.25 ± 0.05	0.5 ± 0.1
2 years old, Kaliningrad, 2021	53.5 ± 0.5	25.1 ± 0.5	19.7 ± 0.5	1.82 ± 0.05	0.3 ± 0.1
8 years old, Penza, 2022	55.5 ± 0.5	20.9 ± 0.5	19.5 ± 0.5	2.63 ± 0.05	0.9 ± 0.1
1 year old, Novosibirsk, 2020	43.2 ± 0.5	19.7 ± 0.5	20.2 ± 0.5	2.95 ± 0.05	0.9 ± 0.1
1 year old, Irkutsk, 2022	44.4 ± 0.5	17.1 ± 0.5	17.9 ± 0.5	2.61 ± 0.05	0.8 ± 0.1

## Data Availability

Not applicable.

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
