# Peer review of "Evaluation of Chemical Composition of Miscanthus × giganteus Raised in Different Climate Regions in Russia"

_plants, 2022, doi:10.3390/plants11202791_

Round 1
Reviewer 1 Report
the subject of the paper is of great interest dealing with plant biomass production, both as an energy source and as a carbon sink. From the results of the present paper, there are two factors affecting cellulose content in Miscanthus plant tissues: climate and age. The other relevant factor is the type of tissue with higher amount of cellulose in stems than in leaves.
There is an important information missing: the biomass production at each of the seven plantation zones from where the samples were collected. An average yield could help or even an estimated production.
Line 104. "Samples were provided by suppliers" : who were they?
lines 105-108. Grouping the samples according to climate conditions in three regions can be ameliorated with more specific information of rain and temperature of the sites.
line 111. The size or the weight of the samples should be included.
I find that this part of the paper, Miscanthus samples, should be improved.
Author Response
The authors’ response to Reviewer 1' comments has been uploaded as a separate PDF file.

Reviewer 2 Report
Summary. The authors report on chemical (lignin, cellulose, pentosans, ash, and extractives) composition of Miscanthus x giganteussampled in different regions of Russia. The stand were of different age. In a subset of samples they analyse the composition separately for leaves and stem. The authors claim that there are not studies on chemical composition (L88). They presnt data for different plant age, regions, and stem leaves. In their discussion they compare their results to some findings of other authors.
It is of general interest to get more detailed information on the biomass crop Miscanthus. While a lot of studies are available there are less studies available for Russia. The results would add to the knowledgebase but it is not adding new findings or insight. Such kind of studies have been conducted before. But the data are of interest for the scientific community.
I have the following concerns about the study which I try to outline in the following. More specific comments could be found in the attached manuscipt (PDF).
Studies conducted in Russia. One could get the impression that this is the first study on Miscanthus in Russia. In the introduction the authors do not make any reference to miscanthus cultivation in Russia or the importance of Miscanthus for Russia as most other scientists do, when they are reporting on cultivation or chemical composition of Miscanthus. I have cited a few in the PDF. What is concerning me most is, that two studies (iam aware of) on the chemical composition of Miscanthus in Novosibirsk are not mention. One could expect that the authors know these studies. Good scientific practice is based on the believe that authros do cite comparable studies. This is mandatory.
Citations. In the introduction I found at least some references which must be considered 2nd order citation. I have commented on this issue in the PDF. This must be revised and is a mandatory request.
Study design. The authors do not give important details on the study. For example weather data or soil data are missing which are important to understand the conditions in the region. The authors mentioned three harvest years (2019- 2021), but it is not clear which sample do belong to which year. The authors seem to ignore the importance of interannual variations of Miscanthus traits. There are basic information on the study missing: where the Miscanthus genotype identical? It is known, that M. x giganteus is not a single genotype and that these genotypes could be accounted for a lot of differences. It is not mentioned, if the sample were all from the same stand (there are contradictionary information given). You will find more comments in the PDF. Material and methods must be revised accordingly giving all information mentioned here or in the PDF.
Reporting. Basic information is missing: there are no yield data, there are no data on weather conditions, there are no data on crop management (e.g. plant density, fertilization). The yield data are mandatory. There is no information given on the stem-leaf ratio (as said before Mxg could be understand as several different genotypes). There is not statistical information given. This is mandatory.
Literature. As stated above, important literature is missing, some references are 2nd order citation. This is basic information the authors must provide. They should not leave it to the reader to find out.
Specific comments. In the PDF I have given more detailed comments which must be considered accroding o the color code describe in the PDF.

Author Response
The authors’ response to Reviewer 2’s comments has been uploaded as a separate PDF file.

Round 2
Reviewer 2 Report
Dear authors,
you responded to all my comments and I found that you have made the changes as requested or you justify why to not change.
Comment. I do disagree to your conclusions why you do not cite the papers on chemical composition of M. sinensis and M. sacchariflorus cultivated in Russia, because: (1) Both are available in English. (2) Miscanthus x giganteus is an allotriploid crossing of M. sinensis and M. sacchariflorus. So, the information on their chemical composition could be discussed here too. With that given you could discuss, if the chemical composition of these two species are different when they are cultivated under Russian conditions in comparison to other climate regions. To broaden the discussion on chemical composition by including this two species would give a much better insight into the discussion, if under Russian climatic conditions the chemical composition is affected in the same way over three species. I agree, that the paper by Baibakova et al. can be omitted, because chemical composition is only mentioned once and just for M. sacchariflorus. I do not see the problem of self-citation here.
Remark. An additional suggestion / idea. I just have noted that you did not include statistical information on significant differences in table 2. It is not critical, but it might help to make comparison easier when you could refere to signgicant differences between: a) region, and / or b) age. Maybe you could do the same for figure 1 too. It would give the reader an information, if the observed differences are signficant or not. But I did not comment on it in the 1st round, I will not make this a request here.
Both comment and remark and should not slow down the process, but I would strongly suggest to include the paper by Dorgina et al.